# GP and parent dissonance about the assessment and treatment of childhood eczema in primary care: a qualitative study

Kingsley Powell,[1] Emma Le Roux,[1] Jonathan Banks,[2] Matthew J Ridd[1]

## ABSTRACT

**Objectives** To compare parents' and clinicians' perspectives on the assessment and treatment of children with eczema in primary care.

**Design** Qualitative interview study with purposive and snowball sampling and thematic analysis.

**Setting** 14 general practices in the UK.

**Participants** 11 parents of children with eczema and 15 general practitioners (GPs) took part in semistructured individual interviews.

**Results** We identified several areas of dissonance between parents and GPs. First, parents sought a 'cause' of eczema, such as an underlying allergy, whereas GPs looked to manage the symptoms of an incurable condition. Second, parents often judged eczema severity in terms of psychosocial impact, while GPs tended to focus on the appearance of the child's skin. Third, parents sought 'more natural' over-the-counter treatments or complementary medicine, which GPs felt unable to endorse because of their unknown effectiveness and potential harm. Fourth, GPs linked poor outcomes to unrealistic expectations of treatment and low adherence to topical therapy, whereas parents reported persisting with treatment and despondency with its ineffectiveness. Consultations were commonly described by parents as being dominated by the GP, with a lack of involvement in treatment decisions. GPs' management of divergent views varied, but avoidance strategies were often employed.

**Conclusions** Divergent views between parents and clinicians regarding the cause and treatment of childhood eczema can probably only be bridged by clinicians actively seeking out opinions and sharing rationale for their approach to treatment. Together with assessing the psychosocial as well as the physical impact of eczema, asking about current or intended use of complementary therapy and involving parents in treatment decisions, the management of eczema and patient outcomes could be improved.

[1]Centre for Academic Primary Care, Bristol Medical School, University of Bristol, Bristol, UK
[2]National Institute for Health Research Collaborations for Leadership in Applied Health Research and Care West (NIHR CLAHRC West), University Hospitals Bristol NHS Foundation Trust, Bristol, UK

**Correspondence to**
Dr Matthew J Ridd;
m.ridd@bristol.ac.uk

and the appearance of the skin. Emollients are usually used in the maintenance phase and different-strength topical corticosteroids (TCS) or calcineurin inhibitors are used to treat an exacerbation or 'flare'. Consequently, treatment regimens can become complex and poor adherence has been associated with poor clinical outcomes.[1]

Factors affecting treatment adherence and outcomes in childhood eczema include: the time taken to apply treatments; confusion about what, when and how to use prescribed medications[2] and parental fear of TCS side effects.[3 4] The doctor–patient relationship is a key factor influencing treatment outcomes.[2] Good verbal and non-verbal communication, shared decision-making, practitioner interest in the patient and acknowledgement of psychosocial issues contribute to a more positive relationship which in turn can lead to improved treatment adherence.[2] Parents' views of and approach to eczema treatments may be different to those of a clinician. In the UK, most children with eczema are managed in primary care yet parents can feel that the condition is not taken seriously enough by their general practitioner (GP).[5] However, research in this area has focused on the views and experiences of parents and children and the perspective of primary care clinicians,

## INTRODUCTION

Eczema is a common childhood condition characterised by dry and itchy skin.[1] However, due to the fluctuating nature of the condition, carers must adapt their treatment regimen according to the child's symptoms

**BMJ**

and particularly the interaction between GP and patient has not been examined in any detail.

As part of a wider study to develop an eczema written action plan (WAP)[6] for children, we explored the experiences of managing eczema from a range of perspectives. In a previous publication, we described the clinical perspective in relation to training, diagnosis and treatment.[7] In this paper, to better understand the issues and their impact on the care of children with eczema more generally, we extend this analysis by comparing for the first time the contrasting views of GPs and parents.

## METHODS

Qualitative interview study to identify the experience of managing childhood eczema in primary care from the perspective of GPs and parents.

### Recruitment and sampling

We interviewed GPs and parents of children under 12 years with eczema, the age group representing the majority of eczema cases in primary care.[8] Parents were recruited from five socioeconomically diverse GP surgeries by written invitation and sampled by the child's age, gender and ethnicity, and self-reported eczema severity.[9] Children were given the opportunity to participate in the interview at their parents' discretion. GPs were recruited from the same practices and through personal and professional networks. They were sampled by gender, years in job role, experience of using and perceived value of WAPs for eczema, and sociodemographic area of the practice. Parents' Index of Multiple Deprivation (IMD) were obtained using their home postal code.[10]

### Data collection

Data collection took place between May 2016 and February 2017. All participants gave written informed consent to take part in the study. Interviews lasted typically between 45 and 60 min, and took place in participants' homes, on National Health Service premises or by telephone. The interviewers (KP, a non-clinical researcher and ELR, a GP with a specialist interest in dermatology) used a topic guide specific for each participant group (ie, parents and GPs) to facilitate discussion. The initial versions were based on the findings of a qualitative and quantitative literature review of self-management/WAPs in children with long-term conditions/eczema.[11 12] In discussion with the research team, they were refined iteratively over the

course of the study, as new areas warranting further exploration emerged. Table 1 shows the issues covered that are relevant to the findings in this paper. ELR interviewed most of the GPs (13/15) and KP interviewed all of the parents. To minimise any biases, both researchers wrote reflective and reflexive accounts throughout the interviews and we discussed any issues with the study team, made up of both clinical and non-clinical researchers, at regular team meetings. KP and ELR also independently double coded a sample of the transcripts (see Data analysis section). Data collection ended once we achieved data saturation within the parent and GP groups with respect to the main research questions around developing a WAP. While the main research question was focused on the WAP development, this was underpinned by identifying issues that parents and children face in primary care consultations and therefore the findings presented in this paper were a key element of our data collection.

### Data analysis

All interview transcripts were transcribed verbatim and imported into QSR NVivo V.10 software. A coding framework, as part of the constant comparative thematic analysis approach,[13] was developed and refined by KP and ELR alongside data collection and agreed with the wider study team. KP and ELR separately applied the coding framework (see online supplementary appendix A) to eight transcripts and discussed and resolved any discrepancies to maximise inter-rater coding agreement. Themes were developed by line-by-line analyses and coding of the data which led to clear patterns in the data from both parents and GPs. Codes were compared within and across participant groups and then grouped into themes and subthemes.

### Patient and public involvement

Between two and four parents met three times in 1.5–2 hours small group meetings over the duration of the broader study. As part of the development of the eczema WAP, they were invited to comment on the findings. They reported that the 'barriers to treatment' described in this paper had strong face validity.

### RESULTS

We interviewed 11 parents (2 with children) and 15 GPs. Participant characteristics are shown in table 2. Of the doctors, 10 were GP partners and 5 associates; two had

| Table 1 Topic guide questions around experience of managing eczema | |
|---|---|
| **Parents** | **General practitioners** |
| ▶ What difficulties have you encountered around diagnosis/ treatment? | ▶ What difficulties have you encountered around diagnosis/ treatment? |
| ▶ What treatments have you used? | ▶ What advice do you give and what do you prescribe? |
| ▶ What advice/support have been given? | ▶ What follow-up arrangements do you make? |
| ▶ What follow-up have you received? | ▶ How do you try and support parents in looking after their child? |

| Table 2 | Participant characteristics | | |
|---|---|---|---|
| | General practitioners | Parents | Child |
| Mean age in years (range) | 43.6 (30–56) | 37.4 (27–48) | 5.3 (0–11) |
| Mean years in role (range) | 14.1 (0.5–29) | – | – |
| Gender | | | |
| Male | 4 | 0 | 4 |
| Female | 11 | 11 | 7 |

a specialist interest in dermatology and two declared a personal experience of eczema (family member affected). Their mean self-reported confidence in managing eczema (0 low, 10 high) was 7.3 (range 5–9). Participating parents' children had a mean eczema severity score of 15.3 (range 1–27), based on the Patient-Oriented Eczema Measure[9] (based on the most affected child for 10 of the 11 parents). Parents had an mean IMD of 5.2 (range 1–9; 1, low, 10 high).

Parents and GPs had different perspectives on the causes, impact and treatment of the condition. We explore the divergent perspectives between parent and GP across several dimensions of primary care consultations for childhood eczema, including assessment, treatment options and treatment adherence/efficacy.

### Divergent views about assessment and cause

One area of division between GPs and parents was around the assessment of the condition and its severity:

I think patients' understanding of what bad eczema is often very different from ours so sometimes they say oh it's a terrible rash and, you know, it's the faintest, palest little bit of redness so because they haven't ever seen the severity of what eczema can be (019G:GP).

In describing the condition, GPs focused more on the physical appearance of eczema and symptoms such as scratching and disturbed sleep. In comparison, parents emphasised the wider psychosocial impact of eczema on the family, social activities and emotional well-being:

there's no way we could even contemplate camping. There's sort of limits- every single night they've got to have a bath, its- he can't have a normal life but don't get me wrong, it's not life threatening and, you know, compared with what he could have its amazing but people have no idea, they haven't got a clue the impact it has. (025P:parent).

Parents felt this wider impact was often overlooked/ underestimated by GPs:

You kind of go in and the doctor thinks oh child's a bit itching, they're not sleeping, rah rah rah, is this just a parent who is being over reactive when in

reality you've got a screaming child who is crawling at their skin (034P:parent).

There were also differing views between parents and GPs about the cause of eczema. Parents often spoke of seeking a 'cure' and held beliefs about an underlying, remediable cause to their child's eczema, most commonly allergy, whereas GPs approached it as an incurable, long-term condition:

when a child is suffering from the eczema the main thing we have to focus on what causes it. It could be for definite it was the diet, it was like allergic to food… (018P:parent)

I don't think they find it difficult to accept that this is called eczema. I think its difficulties accepting that it will be chronic and they will have a tendency to it and accepting that's it not caused by something that they can treat and then it will go away. (021G:GP)

GPs spoke of the challenge of addressing these parental beliefs and resolving discrepant perspectives:

I think they expect that if they can find the one trigger for the eczema they can make it all magically go away so you have to unpick that. (041G:GP).

### Managing divergent views around allergy and complementary/alternative medicine

GPs described how in addition to managing the child's condition, they 'managed' parent views about allergy testing and non-prescribed treatments. GPs noted that parents commonly voiced concerns about allergy during consultations. The difficulty for GPs was weighing up parents' concerns against the clinical likelihood of an allergy, the poor accuracy of the tests and concern about overloading the allergy services:

You take a food history and see if there are any triggers but if they can't usually identify any I don't tend to pursue that. I don't know if that's good practice or not to be honest but it's my general approach to things. (030G:GP).

Strategies to manage this tension varied. Some GPs spoke of acknowledging parents' beliefs ("trying to understand what their concern is" (017G), or taking an allergy focused history to identify a potential problem and, if not, diplomatically explaining that allergy is unlikely to be a factor. Others spoke of trying to "steer" (031G) parents away from this idea and "encourage them to stick to the mainstay" (021G), that is , controlling it with emollients:

I try and explore what they think allergy testing is and what they think they will achieve by having it and ultimately when they- what they're actually asking for is a referral to a specialist which is in some cases entirely fair enough but I think as I said I think they come in armed with a lot of misinformation about it and kind of hope that it's the holy grail which you can

completely understand, but, yeah, I think it's a slightly murky field (041G:GP).

When parents in our study had raised concerns about allergy, they said they were happy to take their GPs' advice. However, they had difficulty reconciling this advice when it conflicted with that received from other sources such as the internet and peers:

> I: 'Is there anything else that would have helped you during that time?
>
> P: 'Yes, to get a concise answer on whether allergy testing is worth doing or not 'cos the number of people who've said to me over the years oh well when I had eczema or my, you know, seventh cousin removed had eczema then they cut out this, that and the other, if I'd followed all of that he would eat nothing at all and yet I've always been told by the medical profession we don't allergy test children with eczema because sometimes they'll react, sometimes they won't…' (025P:parent).

One parent reported that the GP 'dismissed' their concern about allergy:

> My son is allergic to dairy but she [daughter] isn't but I did say to the GP could it be that she shouldn't have dairy and she said- she just dismissed that and said no but I- it's difficult isn't it? I don't know the evidence about, you know, allergies and skin conditions….if there is research about diet and eczema I'd like to know about that (028P:parent).

Parents understood 'complementary and alternative' treatments differently. Some talked about products bought over-the-counter (OTC), such as moisturisers and coconut oil, which was common place; while others talked of herbal remedies and acupuncture, hereon referred to as CAM (complementary and alternative medicine). Some parents turned to CAM in the hope of finding a cure, or in an attempt to better manage eczema symptoms, while for others it was about using more 'natural' products that they deemed to be safer for their child's skin. Parents felt these therapies were effective and they wanted recognition of this from their clinician:

> And also not dismissing, you know, if you are somebody that wants to use more natural products on your child not to dismiss that, you know, as- for instance I use coconut oil for her cradle cap and that's actually been amazing, you know, rather than using all these different shampoos. There are some natural things that really do work and, you know, you kind of want that holistic, you know, care for your child and just that that's accepted in the medical world. (028P:parent).

GPs reported that use of OTC and CAM use among parents was common and uncommon, respectively, although one GP felt "we probably only know the fraction of it" (031G). They felt unable to endorse use of CAM or OTC products because of their unknown effectiveness and concerns about harmful additives:

> Yeah, people talk about lots of other products, aloe vera comes up, but it's difficult for me to assess whether- what the efficacy of those are so I tend to sort of gently, not sort of poo poo it but I gently steer them back to what I know works because I don't always know what additives or product, you know, how good they are really (019G:GP).
>
> I know you're not supposed to use the Chinese things anymore 'cos they've probably got steroids in them aren't they?' (016G:GP)

GPs varied in whether they actively probed parents on their use of CAM or OTC products. One GP said she specifically asked parents, others said parents " tend to come out with it" (035G), whilst for others still it was not "on the top of my list to actively ask" (031G:GP).

Parents and GPs agreed that limited time within the consultation was a contributory factor for these issues and concerns about allergy not being addressed:

> in the ideal world, yes, me and my nurse we would be spending time explaining all of it but in the world we live in that's impossible' (015G:GP)
>
> you only get like ten minutes with the doctor don't you? It's very brief, and they'll say yes its eczema, here's the cream, whack it on two to three times a week and then they send you out the door but you don't know, like you say, it could be an allergy thing that started it off. (027P:parent).

### Treatment decisions, adherence and efficacy

GPs and parents reported a trial and error approach to finding an emollient that suited the family. Eczema guidelines advise healthcare professionals to offer parents and children a choice of emollients.[1 14 15] While several GPs cited the importance of patient preference, their prescribing decisions around emollients seemed to be based on their own preferences or their experiences with other parents, rather than involving parents on an individual basis:

> So I tend to use quite a lot of Hydromol ointment 'cos from my own experience of my daughter found it really good… (035G:GP)

This mirrored the parents' experiences of encounters with GPs, where their role in emollient choice was often depicted as passive compared with the directive manner of the GPs:

> …the GP does his thing and I'm off with my big jars of moisturisers… (033P:parent)
>
> …they're like oh it's just eczema, like you know chuck that at you and I, like I said, that's the second dose now he's had of that one… (014P:parent)

Poor treatment adherence in primary care was commonly reported among GPs. With TCS, they said it

was related to parental fear of their side effects, that is, skin thinning, which was corroborated by the parents themselves. With emollients, some GPs felt that it was a result of parents having unrealistic expectations:

they just want it fixed now, they're not thinking to keep on applying, they'll use it short term and then it'll be ok and then they'll come back. (029G:GP)

I think its difficulties accepting that it will be chronic and they will have a tendency to it and accepting that's it not caused by something that they can treat and then it will go away. (021G:GP).

Conversely, many parents reported persisting with topical therapy, but declared feelings of desperation and frustration at the futility of topical treatments due to their apparent inefficacy:

It just kept on getting worse and worse. I've used everything. I've used Oilatum, Aveeno, I've used derma- is it Dermatol or something like that for the head. I've used steroid creams but then I've just stopped using steroids because I looked at whenever they go- if it goes it comes back when you stop and it's not really advised to use it for a long time anyway so what's the point? (033P:parent)

Parents also described feelings of guilt about requesting more emollient from their GP, or said that requests were met with resistance:

Sometimes I have to fight my corner. I've now got it so that he gets- we get through 500 grams of hydromol a day so at one point people would only give me enough to last barely a week. I've now got it so that we pick up sort of ten kilos a time. Yeah, sometimes-depending on who's doing the repeat prescriptions you sometimes have to fight your corner a bit and I have to make sure I write a long letter to my GP if we're going away for longer than a week or two to explain why I need ridiculous amounts of everything (025P:parent)

I feel a bit guilty asking for more prescriptions at the doctors (laughs) they must think oh I only did that, you know, a month ago or whatever. (028P:parent)

Despondency with topical therapy appeared to relate to parents' lower confidence in clinicians who did not have specialist skills/knowledge. They described more confidence in dermatologists and GPs with a specialist interest or training in dermatology as they felt they gave more credible advice, and treatments that 'worked':

for credibility, to believe what they tell- what they tell me, what they tell me works and I'm going to follow it I would want it to be someone who is trained by dermatology or Eczema Society and does it frequently for other families. (000P:parent)

I think she just gave me a bit more information really and I trusted her experience of what is actually going to work … and she explained that eczema quite frequently gets infected and it would be better to use that (Dermol) regularly as her moisturiser (28P:parent).

## DISCUSSION
### Summary
We identified several areas of divergent views between parents and GPs regarding disease aetiology, assessment and treatment. First, parents looked for a 'cure' for eczema, often fixated on the role of allergy. Clinicians, on the other hand, were more focused on just 'managing' the condition. Second, some parents wanted clinicians to provide a more holistic approach to assessment and treatment, and to acknowledge the role of complementary or non-prescribed therapies. However, clinicians did not regularly enquire about use of non-prescribed treatments and were not willing to support their use because of uncertainty about their provenance and absence of evidence regarding their efficacy or safety. Third, parents perceived GPs as dismissive and felt they did not recognise the full impact of eczema on the child's quality of life. GPs felt that poor response to treatment was a result of lack of adherence to emollients, which was linked to parents wanting a quick 'fix'. However, parents reported generally persisting with topical therapy but often felt it was ineffective. Finally, parents described GP–parent interactions as being dominated by the doctor, and the lack of parental involvement in treatment decisions was apparent in data from both perspectives.

### Strengths and limitations
To our knowledge, this is the first qualitative study to directly explore and compare GP and parent perspectives on eczema management and treatment. Our sample gave us a diverse range of views from parents (with respect to age, ethnicity of child and their self-rated eczema severity score) and GPs (with respect to age, years of working and sociodemographic population of the practice). However, GPs were predominantly female/GP partners and several had a professional or personal interest in eczema meaning the views captured may not be representative of all GPs. Despite this, our GP participants appeared to be unaware of the importance parents place on the wider impact of eczema, for example, the emotional and psychosocial elements. While some of the GPs and parents came from the same practice, the interviews were not 'paired'. Paired interviews may have added to the richness of the data and our understanding of areas of divergence. The data may also have been influenced by the status of the interviewer; the majority of interviews with GPs were carried out by a GP researcher with a specialist interest in dermatology and the parent interviews by a non-clinical researcher.[16] Data gathered from the two children interviewed alongside their parents are not presented because they did

not contribute any information around GP–patient dissonance.

## Comparison with other literature

Patient resistance to topical treatments; perceived GP disinterest in eczema and its psychological impact; concerns about GP competency; and parental pursuit of allergy as cause and a 'cure' have been previously described.[5 17 18] Gore et al[19] outlined parents' wide ranging and complex informational needs, including the requirement to address concerns about aetiology and any role for diet. However, we also found differences in parents' and GPs' approaches to assessing eczema, their views of the reasons for barriers to effective emollient therapy and their perspectives around CAM use.

Parents of children with eczema often want an 'active' role in treatment decision-making in order to address a perceived lack of (or inconsistent) information and the lack of time and/or interest from health professionals.[19] Patients' agendas are commonly not voiced which can lead to misunderstanding.[20] Showing empathy and asking questions about the impact of the condition can positively affect the relationship between parent and GP,[2 17] and is in line with current clinical guidance which advocates a holistic approach to eczema assessment.[1 14] However, our study shows that even when GPs are aware of disparate views, for example, parent beliefs around allergy, it does not necessarily result in discussion and resolution of these beliefs. Rather, GPs described a tendency to avoid rather than confront them, with concerns about the access to, and interpretation of, allergy tests cited as factors.

Previous research suggests that CAM use is common among patients with eczema.[21 22] Despite the risks attached to the use of some forms of CAM,[23] it is promoted as being effective and safe,[24] and is used by patients looking for 'natural' treatments and a 'cure',[22 25] both of which were reasons for its use by parents in our study. Clinical guidelines encourage open discussions between healthcare professionals and parents about CAM treatments for eczema and explanations about the lack of evidence for its safety and efficacy.[1 14] However, patients rarely disclose CAM use[26] and healthcare professionals rarely ask about it.[27 28] Similarly in our study, conversations about CAM were not high on the GPs' agenda, which may partly be due to their perception that it is not commonly used for eczema.

Another reported reason for CAM use is patient dissatisfaction with conventional eczema treatments,[22] where parents sometimes feel emollients do not 'work'.[5] This was reflected in our interviews with parents where they persisted with topical therapy but often felt it was ineffective. Santer et al[29] found that parents had mixed views about long-term use of emollients to prevent flare-ups and this was echoed by the GPs in our study who said that parents were not using emollients for long enough to be effective and were expecting results with short-term use. Education about the rationale for emollients may lead to more positive parental attitudes towards long-term use.[29] Emollients are widely accepted in the medical world as the mainstay for eczema management in that they are the first line of treatment even when the skin is clear.[30] They come in different formulations, but there is limited evidence to support the use of one emollient over another, therefore patient preference is a key part of prescribing decisions.[31] However, the other key treatment in primary care for children with all but the mildest eczema is appropriate use of TCS, concerns about which can be a significant barrier to achieving and maintaining disease control.[5 32]

## Implications for research and practice

Good eczema care requires a significant degree of self-management but engagement in such processes needs to be built on a solid foundation of understanding around causes and treatment between GP and parent. Rather than shying away from these conversations, clinicians should be prepared to engage in these discussions and explain the rationale behind their approach. Longer, initial consultations may facilitate this, although this must be accompanied by skills and knowledge that builds parental trust. There is some evidence that nurse-led clinics may improve eczema control which could be used to support time-limited GPs, although evidence of benefit in a primary care setting is lacking.[11]

Clinicians are reminded that assessments of disease severity should include questions about psychosocial impact as well as a physical examination. Better shared decision-making, particularly around treatment preferences, together with signposting of reliable sources of information may also improve treatment adherence. As evidence to guide emollient choice and the long-term safety of TCS is lacking,[33] healthcare professionals and parents need to openly discuss individual needs and preferences to find a treatment that is both suitable and acceptable for the family. Treatment discussions should also include current or intended use of CAM as parents may be unaware of its potential risks. Understanding why a parent is using CAM, for example because they are dissatisfied with their emollient, may inform future prescribing decisions.

Our work, and that of others, could be extended through observation of consultations, to assess how these divergent perspectives are played out in practice and how they might be addressed. In addition to supporting GP undergraduate and postgraduate dermatology education, research is needed to find ways to support consultations in primary care for eczema. The WAP, we have developed as part of the wider project, may be one way of facilitating these discussions.[6]

**Twitter** @riddmj @emmaleroux12 @jonbanks10 @apachestudy

**Acknowledgements** We would like to thank all the participants who participated in this research, the West of England Clinical Research Network; the UK Dermatology Clinical Trials Network; the practices who were involved in facilitating this study and the members of the Patient and Public Involvement and Engagement group. Thanks also to Alice Malpass, University of Bristol, who commented on the study protocol; Nancy Horlick, University of Bristol,

who transcribed the audio files and to the peer reviewers of this paper for their thoughtful comments.

**Contributors** KP is the main author, leading on data collection, analysis and write-up. ELR was also involved in data collection, analysis and contributed to revisions of the write-up. MJR obtained funding for the study and MJR/JB were involved in the methodological design and contributed to analysis via discussions during regular research meetings, as well as having significant input to drafts of the written paper.

**Funding** This work/MJR time was supported by the National Institute for Health Research Postdoctoral Research Fellowship (PDF-2014-07-013). KP was funded by National Institute for Health Research (NIHR)'s Research Capability Funding from the Avon Primary Care Research Collaborative on behalf of Bristol, North Somerset and South Gloucestershire CCGs. ELRs time was supported by Elizabeth Blackwell Institute for Health Research, University of Bristol and the Wellcome Trust Institutional Strategic Support Fund. JB time was supported by the NIHR Collaboration for Leadership in Applied Health Research and Care West (CLAHRC West) at University Hospitals Bristol NHS Foundation Trust.

**Disclaimer** The views expressed in this publication are those of the authors and not necessarily those of the NHS, the National Institute for Health Research or the Department of Health.

**Competing interests** None declared.

**Patient consent** Detail has been removed from this case description/these case descriptions to ensure anonymity. The editors and reviewers have seen the detailed information available and are satisfied that the information backs up the case the authors are making.

**Ethics approval** The study received ethical approval from the Yorkshire and the Humber—Bradford Leeds Research Ethics Committee (REC reference 16/YH/0179).

**Provenance and peer review** Not commissioned; externally peer reviewed.

**Data sharing statement** Participants in this study consented to other researchers using the anonymised data to support future research, the data of this study being used to promote scientific knowledge and for no other purpose than research. Therefore, for bona fide researchers, the fully anonymised dataset has been deposited at the University of Bristol Research Data Repository (http://data.bris.ac.uk). A metadata record is openly available by the repository, with a link (data-bris@bristol.ac.uk) to the Research Data team at Bristol who provide information on how data can be accessed by bona fide researchers, and who will assess the motives of potential data reusers before granting access to the data. No authentic request for access will be refused and reusers will not be charged for any part of this process.

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
