## [Reviewer comments · BMJ Open]

ARTICLE DETAILS

TITLE (PROVISIONAL)	GP and parent dissonance about the assessment and treatment of childhood eczema in primary care: a qualitative study
AUTHORS	Powell, Kingsley; Le Roux, Emma; Banks, Jonathan; Ridd, Matthew

VERSION 1 – REVIEW

REVIEWER	Fiona Cowdell Birmingham City University UK
REVIEW RETURNED	21-Sep-2017

GENERAL COMMENTS	This is an interesting and worthwhile paper which could inform eczema treatment in primary care. It's generally well written but the information regarding qualitative methodology lacks detail at present. Specific points are outlined below Abstract - add in primary care to objectives, add type of qualitative study and how many practices involved line 21 - is 'low' unrealistic expectations an error? Method section - needs much more detail about the type of qualitative research method used, and the preconceived ideas of the researchers (particularly the GP with an interest in dermatology) and how biases were managed. Need to state how topic guide was developed. The relationship between this study and the WAP development needs to be specified. Again a bit more detail - can you provide a figure showing the coding framework and explain how this was devised and how themes were developed It would be helpful to name the themes in the section on data analysis Did you consider convergence as well as dissonance? It's a shame that the child participants views are simply discounted Table 1 - would have been interesting to consider what has been good as well as difficulties
---

REVIEWER	Gayle Fischer University of Sydney Australia
REVIEW RETURNED	22-Sep-2017

GENERAL COMMENTS	I agree with all of your findings which reflect my own experience with managing eczema of all degrees of seriousness as a dermatologist. I note in your discussion you state that emollients are the mainstay of treating eczema. However as far as prescribed medication goes, this is a statement that also ought to be made regarding topical corticosteroids. If it is the standard in your area of general practice to rely on emollients for managing eczema, then it is no surprise that parents are frustrated with lack of response to treatment. In all but the most trivial cases (and this can surely not be said of a child whose eczema is so severe that it impacts quality of life) topical corticosteroids must be included. The perception by both parents and unfortunately sometimes GP's is that these medications are much more dangerous than they really are. As a result they are underused. The parental statement regarding "what's the point because the result is temporary and you can't use them longterm" typifies this. Can you address this in your discussion? At the heart of helping parents to manage their children, in my opinion, is guiding them to use medication confidently. This means empowering them to resist a great deal of popular advice particularly around the dangers of topical corticosteroids, the relevance of allergy and the relentless search for a cure. The latter is a natural response to any diagnosis and self-directed actions supported by information on the internet and CAM are more likely to happen if medical treatment is not effective. I agree that parents are more likely to relinquish unrealistic expectations if they are listened to and validated. However they must also be given treatment that will work effectively and quickly as well as the reassurance that it is as safe as natural therapy. Endorsing the latter is not necessary and is dishonest if you as a doctor wish to rely on evidence based treatment. It can be acknowledged, yes, but your role is to convince that medical treatment works best. The other issue you mention is lack of time. If 10 minutes is all the time GPs have to spend per consultation there is little chance of achieving your goal of better understanding between GP and patient. The GP has barely enough time to make the diagnosis and explain the treatment in this time. Where do nurses fit in supporting the goal of better communication? There are references on nurse lead clinics you could reference in your discussion.
--

REVIEWER	Dr Fiona Collier Dermatology Department, Stirling Community Hospital, Scotland and Alva Medical Practice, Alva, Scotland.
REVIEW RETURNED	05-Oct-2017

GENERAL COMMENTS	This study used qualitative methodology, in the form of semi-structured interviews, to explore and contrast the perceptions and expectations of parents of children with eczema, and those of GPs who treat them. The subject matter is important and has been recognised as research priority by NICE (1). Its inclusion of GPs' viewpoint is novel, with neither a literature search, nor a search of
---

the Centre for Evidence-based Dermatology's collection of eczema systematic reviews, eliciting any similar studies (2) (3).

The qualitative methods were suited for an exploratory study, and the setting was appropriate in a variety of GP surgeries. The topic guides and participants' quotations gave a good flavour of the interviews.

The abstract accurately reflects the study design, and conclusions were supported by themes identified from the 'constant comparison' method. The method of analysis, and how quality was assured in this process, were clearly described. Ethical approval for the study was stated, and sources of funding for the research were clearly described. References were relevant and reasonably current. The authors also completed a COREQ statement designed to highlight issues reflecting quality in qualitative research (4).

There were a few points that it would be helpful for the authors to clarify. In the table of participants' characteristics, it would be helpful to know the socio-economic class and ethnicity of parents, if available, as these are both factors that can affect the presentation and management of eczema (5) (6). This would also allow readers to assess applicability of the findings to their own situation. Although it is stated that a range of socio-economic areas were represented in the GP practices, this does not necessarily mean that there is a similar range of patients' socio-economic backgrounds. It would have been interesting to know if there were any correlations between different viewpoints and characteristics of participants, for both GPs and parents. As the authors mention, linking parent views with those of the GP seeing that particular child might also have added richness to the data, as might seeking validation by participants of the themes identified.

The contribution of the local Patient Public Involvement Group is mentioned, and it would be useful to have some more details of what this involvement was, and how it contributed to the research design.

Overall this was a very well-planned and meticulously-executed qualitative study, conducted in a real world setting, and addressing an important topic.

References

1. Lewis-Jones S, Cork M, Clark C. Atopic Eczema in Children. National Collaborating Centre for Women and Children's Health, report on behalf of National Institute for Health and Clinical Excellence pp20-21 pub RCOG Press 2007
2. Centre for Evidence –Based Dermatology systematic reviews in eczema .Available at:
<http://www.nottingham.ac.uk/research/groups/cebd/resources/eczema-a-systematic-reviews.aspx>. Accessed 05/10/2017
3. Ridd MJ, King AJL, Le Roux E, Waldecker A, Huntley AL. Systematic review of self-management interventions for people with eczema. *Br J Dermatol*. 2017 Sep; 177(3):719-734. doi: 10.1111/bjd.15601. Epub 2017 Aug 2.
4. Tong A., Sainsbury P, Craig J. Consolidated criteria for reporting qualitative research (COREQ). A 32 item checklist for interviews and focus groups. *International Journal for Quality in Health Care*; (19), Number 6: pp. 349–357 10.1093/intqhc/mzm042
5. Charrow A, Xia FD, Joyce C, Mostaghimi A. Diversity in Dermatology Clinical Trials: A Systematic Review. *JAMA Dermatol*. 2017 Jan 4. doi: 10.1001/jamadermatol.2016.4129. [Epub ahead of print]
6. Uphoff E, Cabieses B, Pinart M, Valdés M, Antó JM, Wright J A

	systematic review of socioeconomic position in relation to asthma and allergic diseases. Eur Respir J. 2015 Aug; 46 (2):364-74. doi: 10.1183/09031936.00114514. Epub 2014 Dec 23.
--	---

REVIEWER	Miriam Santer Associate Professor of Primary Care Research University of Southampton UK I have carried out research on this topic, some of which has been referred to in this manuscript.
REVIEW RETURNED	06-Oct-2017

GENERAL COMMENTS	Thank you very much for asking me to review this interesting and well-argued paper. I enjoyed reading it very much. My only difficulty is in assessing the original contribution made here, as the paper refers to 2 other papers from the same dataset that have been submitted or accepted for publication elsewhere (references 6 and 7). Abstract This is well-written, except line 21 where 'low unrealistic expectations' doesn't quite make sense. In line 28 the authors state 'Divergent views between parents and clinicians regarding the aetiology and treatment...'. While the divergent views on treatments are well demonstrated, I was less clear what they meant by aetiology here – if referring to the search for a cure through dietary restriction it might be clearer to say this. Methods Data saturation is mentioned in relation to the main aims of the research but not in relation to the themes presented in this paper. It needs to be clarified whether the authors felt the themes presented here were saturated. Results The authors could be more careful in acknowledging that these are accounts, e.g. Page 6 line 16 "parents... were generally happy to take their GP's advice." What we actually know from this data is that parents said they were happy to take their GP's advice. Discussion It is striking that there is much more data in this paper on views around CAM and emollients than there is on diet/allergy or topical corticosteroids, whereas the latter two have been prominent in existing qualitative research around childhood eczema, only some of which is discussed here. It would have been particularly interesting to contrast parent and doctor views on topical corticosteroids. I wondered whether the authors had reflected on whether the difference in emphasis was due to the nature of their sample, or possibly that data saturation on views of treatment was not achieved in this relatively small sample, or whether this data is presented in one of the other publications from this dataset. Qualitative papers that have identified the importance of parental views on topical steroids include (Jackson et al., 2014, Santer et al., 2012, Smith et al., 2010) Others have found less frustration with emollient therapy than views presented here (Santer et al., 2016)
--

	The conclusions focus very much on clinicians spending more time and carrying out 'better shared decision making'. While this would certainly be useful, it may help to highlight other issues, such as the need for signposting towards reliable sources of information and possibly reflecting on the role of treatments themselves, for instance the uncertainties in parents' minds regarding safe topical corticosteroid use and perceived ineffectiveness of emollients, both of which are areas where the evidence base is not particularly strong. Other explorations of treatment adherence have called for less emphasis on the doctor-patient relationship and a focus on the treatments themselves (Pound et al., 2005) COREQ The COREQ statement does not answer question 13 regarding how many people refused to participate, in that there is no mention of how many invitations were sent. There is inconsistency between the COREQ and Methods section in the paper regarding duration of interviews. JACKSON, K., ERSSER, S. J., DENNIS, H., FARASAT, H. & MORE, A. 2014. The Eczema Education Programme: intervention development and model feasibility. J Eur Acad Dermatol Venereol, 28, 949-56. POUND, P., BRITTEN, N., MORGAN, M., YARDLEY, L., POPE, C., DAKER-WHITE, G. & CAMPBELL, R. 2005. Resisting medicines: a synthesis of qualitative studies of medicine taking. Soc Sci Med, 61. SANTER, M., BURGESS, H., YARDLEY, L., ERSSER, S., LEWIS-JONES, S., MULLER, I., HUGH, C. & LITTLE, P. 2012. Experiences of carers managing childhood eczema and their views on its treatment: a qualitative study. Br J Gen Pract, 62, e261-7. SANTER, M., MULLER, I., YARDLEY, L., LEWIS-JONES, S., ERSSER, S. & LITTLE, P. 2016. Parents' and carers' views about emollients for childhood eczema: qualitative interview study. BMJ open, 6, e011887. SMITH, S. D., HONG, E., FEARN, S., BLASZCZYNSKI, A. & FISCHER, G. 2010. Corticosteroid phobia and other confounders in the treatment of childhood atopic dermatitis explored using parent focus groups. Australas J Dermatol, 51, 168-74.
--	---

VERSION 1 – AUTHOR RESPONSE

Reviewer 1 (Fiona Cowdell)

Reviewer's comments Authors' response

1 Abstract - add in primary care to objectives, add type of qualitative study and how many practices involved

Line 21 - is 'low' unrealistic expectations an error?

Response: We have added in primary care to the objectives section of the abstract, added the number of GP practices involved and removed 'low' which was an error – thank you for highlighting.

2 Method section - needs much more detail about the type of qualitative research method used, and the preconceived ideas of the researchers (particularly the GP with an interest in dermatology) and how biases were managed. Need to state how topic guide was developed.

Response: We recognise this point and have added the relevant detail to the methods section.

3 Methods - Again a bit more detail - can you provide a figure showing the coding framework and explain how this was devised and how themes were developed
It would be helpful to name the themes in the section on data analysis

Response: We have added more detail to the methods regarding the coding framework, how it was devised and how the themes were developed (page 4, 'Data analysis').
We have provided a table with the codes, which could be added as an appendix to the published paper, if the editor wishes to include this.
The main themes are all identified in the results and we do not feel it is necessary to include them in the methods as they were not developed a priori but emerged inductively from the data.

4 The relationship between this study and the WAP development needs to be specified.

Response: This had already been addressed in the last paragraph of the introduction, which we have edited for clarity. (Page 3, lines 75-76)

5 Did you consider convergence as well as dissonance? It's a shame that the child participants views are simply discounted

Response: Thank you for raising this point - we looked for areas of convergence (e.g. parental fear of topical corticosteroids) as well as divergence, but the dominant "overarching theme" was dissonance and that is why, as a novel finding, it is the focus of this paper. As we discuss in the paper, dissonance rather than convergence is likely to have a negative impact on eczema treatment. The views of the child's participants were given full consideration but did not contribute any significant insights.

6 Table 1 - would have been interesting to consider what has been good as well as difficulties Our topic guides were informed by previous research, which identified several problems relating to primary care consultations, hence the initial focus on difficulties. However, all the other areas were explored in a neutral manner, allowing for areas of agreement (such as parental concern about use of topical corticosteroids) to emerge.

Reviewer 2 (Gayle Fischer)

Reviewer's comments Authors' response

1 Discussion - I note in your discussion you state that emollients are the mainstay of treating eczema. However as far as prescribed medication goes, this is a statement that also ought to be made regarding topical corticosteroids. If it is the standard in your area of general practice to rely on emollients for managing eczema, then it is no surprise that parents are frustrated with lack of response to treatment. In all but the most trivial cases (and this can surely not be said of a child whose eczema is so severe that it impacts quality of life) topical corticosteroids must be included. The perception by both parents and unfortunately sometimes GP's is that these medications are much more dangerous than they really are. As a result they are underused. The parental statement regarding "what's the point because the result is temporary and you can't use them long term" typifies this. Can you address this in your discussion?

Response: We agree that topical corticosteroids are a key component to eczema treatment. By 'mainstay', we mean that emollients are the first line of treatment to be used daily even when the skin is 'clear' (NICE, 2007), which we have added to the sentence referring to this in the discussion (page 10, line 352). The consensus in this study was that that topical corticosteroids are underused by parents, and this has been corroborated by other studies. We have changed the final sentence of the "Comparison with other literature" section to reflect this point. (Page 10, line 355-359)

2 At the heart of helping parents to manage their children, in my opinion, is guiding them to use medication confidently. This means empowering them to resist a great deal of popular advice particularly around the dangers of topical corticosteroids, the relevance of allergy and the relentless search for a cure. The latter is a natural response to any diagnosis and self-directed actions supported by information on the internet and CAM are more likely to happen if medical treatment is not effective.

Response: We recognise the point being made here about the impact of popular discourses on use of topical corticosteroids. However, we feel that it is beyond the scope of our data and results to discuss these issues any further in this paper.

3 I agree that parents are more likely to relinquish unrealistic expectations if they are listened to and validated. However they must also be given treatment that will work effectively and quickly as well as the reassurance that it is as safe as natural therapy. Endorsing the latter is not necessary and is dishonest if you as a doctor wish to rely on evidence based treatment. It can be acknowledged, yes, but your role is to convince that medical treatment works best.

Response: As above, we recognise the points being made here but they are beyond the scope of this paper to incorporate.

4 The other issue you mention is lack of time. If 10 minutes is all the time GPs have to spend per consultation there is little chance of achieving your goal of better understanding between GP and patient. The GP has barely enough time to make the diagnosis and explain the treatment in this time. Where do nurses fit in supporting the goal of better communication? There are references on nurse lead clinics you could reference in your discussion.

Response: Thank you for raising this issue. We have added the following sentence to the discussion to incorporate this: "There is some evidence that nurse led clinics may improve eczema control, which could be used to support time-limited GPs, although evidence of benefit in a primary care setting is lacking." (page 10, line 366-368).

Reviewer 3 (Fiona Collier)

Reviewer's comments Authors' response

1 In the table of participants' characteristics, it would be helpful to know the socio-economic class and ethnicity of parents, if available, as these are both factors that can affect the presentation and management of eczema (5) (6). This would also allow readers to assess applicability of the findings to their own situation. Although it is stated that a range of socio-economic areas were represented in the GP practices, this does not necessarily mean that there is a similar range of patients' socio-economic backgrounds. It would have been interesting to know if there were any correlations between different viewpoints and characteristics of participants, for both GPs and parents.

Response: We only recorded the ethnicity of the child, not the parent and therefore we are unable to include this.

We have calculated the average Index of Multiple Deprivation score and added this to the results section, page 5, line 133.

2 As the authors mention, linking parent views with those of the GP seeing that particular child might also have added richness to the data, as might seeking validation by participants of the themes identified.

Response: The contribution of the local Patient Public Involvement Group is mentioned, and it would be useful to have some more details of what this involvement was, and how it contributed to the research design. Unfortunately we were unable to ask participants to reflect on the themes that we identified, but our PPI group (comprising parents of children with eczema) agreed with the themes that emerged from our work developing the WAP as “barriers to treatment”, which we report here. We have added a “PPI” sub-heading to the methods to describe the group’s composition and involvement in the study: ‘Between two and four parents met three times in 1.5-2 hour small group meetings over the duration of the broader study. As part of the development of the eczema WAP, they were invited to comment on the findings. They reported that the “barriers to treatment” described in this paper had strong face validity.’

Reviewer 4 (Miriam Santer)

Reviewer’s comments Author’s response

1 Thank you very much for asking me to review this interesting and well-argued paper. I enjoyed reading it very much. My only difficulty is in assessing the original contribution made here, as the paper refers to 2 other papers from the same dataset that have been submitted or accepted for publication elsewhere (references 6 and 7).

Response: As described in the introduction, this paper arose from a project developing a Written Action Plan for children with eczema. As part of this study, we set out to specifically explore the GPs’ experience and confidence in managing the condition in this population. Two papers from this work are in press (references 6 & 7), the first focusing on acceptability, content and format of a written action plan; the second on GP’s perspectives. However, while some of the findings in the second paper overlap GP views reported in this paper, this paper is the first publication to specifically compare and contrast the views of parents and doctors in a UK primary care setting. We report novel information around dissonance between parents and GPs which is not covered in depth in either of the other two papers.

2 Abstract - This is well-written, except line 21 where ‘low unrealistic expectations’ doesn’t quite make sense.

Response: Please see our response to reviewer 1, response 1 with reference to line 21 of the abstract – this error has been amended.

3 Abstract - In line 28 the authors state ‘Divergent views between parents and clinicians regarding the aetiology and treatment...’. While the divergent views on treatments are well demonstrated, I was less clear what they meant by aetiology here – if referring to the search for a cure through dietary restriction it might be clearer to say this.

Response: We have replaced ‘aetiology’ with ‘cause’ to simplify this. Parents often looked for a remediable cause such as an allergy as explained in the results section.

4 Methods - Data saturation is mentioned in relation to the main aims of the research but not in relation to the themes presented in this paper. It needs to be clarified whether the authors felt the themes presented here were saturated.

Response: The reviewer raises an important point. When we discuss data saturation we refer to the fact that there were no new issues coming up in the interviews. We agree that further exploration is probably warranted to substantiate and deepen our understanding of the issues. We have added the following sentence to the ‘data collection’ section of the methods (page 4, line 107-109): ‘Whilst the main research question was focused on the WAP development, this was underpinned by identifying

issues that parents and children face in primary care consultations and therefore the findings presented in this paper were a key element of our data collection.'

5 Results - The authors could be more careful in acknowledging that these are accounts, e.g. Page 6 line 16 "parents... were generally happy to take their GP's advice." What we actually know from this data is that parents said they were happy to take their GP's advice. We acknowledge this advice and we have changed this sentence to: "When parents in our study had raised concerns about allergy, they said they were happy to take their GPs' advice."

Response: We have also checked the paper for/clarified any other ambiguous statements.

6 Discussion - It is striking that there is much more data in this paper on views around CAM and emollients than there is on diet/allergy or topical corticosteroids, whereas the latter two have been prominent in existing qualitative research around childhood eczema, only some of which is discussed here. It would have been particularly interesting to contrast parent and doctor views on topical corticosteroids. I wondered whether the authors had reflected on whether the difference in emphasis was due to the nature of their sample, or possibly that data saturation on views of treatment was not achieved in this relatively small sample, or whether this data is presented in one of the other publications from this dataset. Qualitative papers that have identified the importance of parental views on topical steroids include (Jackson et al., 2014, Santer et al., 2012, Smith et al., 2010) Others have found less frustration with emollient therapy than views presented here (Santer et al., 2016)

Response: With regards to data saturation, we have now clarified in the methods section that data saturation was reached for the main research questions and that the findings we present here were key building blocks to these research questions. However, further exploration is needed and this may be one reason for the difference in emphasis that the reviewer has noticed. However, another reason for this is that we do not present all the data relating to CAM/diet/corticosteroids in this paper but specifically focus on areas of dissonance.

For example, parents spoke in depth about topical corticosteroids but in terms of dissonance there was not much to report because both parents and GPs agreed that there was steroid phobia among parents.

The views around topical corticosteroids are referred to in this paper, but as stated in the results section, this is one of the few areas where both GPs and parents agreed that there is fear of using steroids among parents and therefore they are underused. As explained in response to reviewer 1, while we sought to identify and explore areas of agreement and difference, the majority of findings fell into the second camp, leading us to focus on the issue of dissonance.

Thank you for highlighting these references, we have incorporated some into the discussion. (Page 10, line 358).

8 The conclusions focus very much on clinicians spending more time and carrying out 'better shared decision making'. While this would certainly be useful, it may help to highlight other issues, such as the need for signposting towards reliable sources of information and possibly reflecting on the role of treatments themselves, for instance the uncertainties in parents' minds regarding safe topical corticosteroid use and perceived ineffectiveness of emollients, both of which are areas where the evidence base is not particularly strong. Other explorations of treatment adherence have called for less emphasis on the doctor-patient relationship and a focus on the treatments themselves (Pound et al., 2005)

Response: We thank the reviewer for these thoughtful comments. We have added the importance of signposting reliable sources of information/the lack of long-term data on the safety of topical corticosteroids to the “Implications for research and practice” paragraph. (Page10, 371-373)

9 COREQ - The COREQ statement does not answer question 13 regarding how many people refused to participate, in that there is no mention of how many invitations were sent. There is inconsistency between the COREQ and Methods section in the paper regarding duration of interviews.

Response: We have added in the number of invitation letters sent to parents, but the number of GPs who received the email asking them to participate is unknown as we asked the research leads to distribute the invitation to their practice, and other GPs heard about the study by word of mouth or professional networks.

Thank you for pointing out the inconsistency of the interview lengths. The paper states the average time, but there were a couple of anomalies outside of this which lasted quite a bit shorter or longer. We have added ‘typically’ to the methods section of the paper.

We hope the changes we have made are satisfactory.

VERSION 2 – REVIEW

REVIEWER	Fiona Cowdell Birmingham City University, UK
REVIEW RETURNED	11-Dec-2017

GENERAL COMMENTS	An interesting and well written paper thank you.
--

REVIEWER	Gayle Fischer The University of Sydney Australia
REVIEW RETURNED	06-Dec-2017

GENERAL COMMENTS	Thanks for your revisions, the paper reads well and is consistent with the findings of other researchers in this field.
---

REVIEWER	Fiona Collier GP Alva Health Centre, Alva, Clackmannanshire and GPWSI Dermatology NHS Forth Valley, Scotland.
REVIEW RETURNED	03-Dec-2017

GENERAL COMMENTS	This is a rigorously-conducted and well- reported qualitative study. The amendments have appropriately addressed reviewers' concerns. Just one further point, the last sentence in the ' strengths and limitations' paragraph could elaborate a little on what themes they are referring to.
---

REVIEWER	Miriam Santer University of Southampton, UK
REVIEW RETURNED	07-Dec-2017

GENERAL COMMENTS	Thank you for addressing my comments on the previous version.
---